# Feasibility study of mitigation and suppression strategies for controlling COVID-19 outbreaks in London and Wuhan

**Po Yang**[1,2]*, **Jun Qi**[3,4]*, **Shuhao Zhang**[5], **Xulong Wang**[5], **Gaoshan Bi**[5], **Yun Yang**[5]*, **Bin Sheng**[6], **Geng Yang**[2]

**1** Department of Computer Science, The University of Sheffield, Sheffield, United Kingdom, **2** The State Key Laboratory of Fluid Power and Electronic Systems, School of Mechanical Engineering, Zhejiang University, China, **3** Department of Engineering Science, University of Oxford, Oxford, United Kingdom, **4** Department of Computer Science and Software Engineering, Xi'an Jiaotong-Liverpool University, Suzhou, China, **5** School of Software, Yunnan University, Yunnan, China, **6** Department of Computer Science and Engineering, Shanghai JiaoTong University, Shanghai, China

* po.yang@sheffield.ac.uk (PY); ruth1012@live.com (JQ); yangyun@ynu.edu.cn (YY)

## Abstract

Recent outbreaks of coronavirus disease 2019 (COVID-19) has led a global pandemic cross the world. Most countries took two main interventions: suppression like immediate lockdown cities at epicenter or mitigation that slows down but not stopping epidemic for reducing peak healthcare demand. Both strategies have their apparent merits and limitations; it becomes extremely hard to conduct one intervention as the most feasible way to all countries. Targeting at this problem, this paper conducted a feasibility study by defining a mathematical model named SEMCR, it extended traditional SEIR (Susceptible-Exposed-Infectious-Recovered) model by adding two key features: a direct connection between Exposed and Recovered populations, and separating infections into mild and critical cases. It defined parameters to classify two stages of COVID-19 control: active contain by isolation of cases and contacts, passive contain by suppression or mitigation. The model was fitted and evaluated with public dataset containing daily number of confirmed active cases including Wuhan and London during January 2020 and March 2020. The simulated results showed that 1) Immediate suppression taken in Wuhan significantly reduced the total exposed and infectious populations, but it has to be consistently maintained at least 90 days (by the middle of April 2020). Without taking this intervention, we predict the number of infections would have been 73 folders higher by the middle of April 2020. Its success requires efficient government initiatives and effective collaborative governance for mobilizing of corporate resources to provide essential goods. This mode may be not suitable to other countries without efficient collaborative governance and sufficient health resources. 2) In London, it is possible to take a hybrid intervention of suppression and mitigation for every 2 or 3 weeks over a longer period to balance the total infections and economic loss. While the total infectious populations in this scenario would be possibly 2 times than the one taking suppression, economic loss and recovery of London would be less affected. 3) Both in Wuhan and London cases, one important issue of fitting practical data was that there were a portion (probably 62.9% in Wuhan) of self-recovered populations that were asymptomatic or

**Data Availability Statement:** We have made all code and data available online, and uploaded relevant supplemental materials. All data and code required to reproduce the analysis are available

online at: https://github.com/TurtleZZH/Feasibility-Study-of-Mitigationand- Suppression-Intervention-Strategies-for-Controlling-COVID-19.git

**Funding:** This work has been supported by WorldWide Universities Network Special Grant Scheme at University of Sheffield, in part by the Open Foundation of the State Key Laboratory of Fluid Power and Mechatronic Systems under Grant No: GZKF-201802, in part by the the National Natural Science Foundation of China under Grant No: 61876166 and 61663046.

**Competing interests:** The authors have declared that no competing interests exist.

mild symptomatic. This finding has been recently confirmed by other studies that the sero-prevalence in Wuhan varied between 3.2% and 3.8% in different sub-regions. It highlights that the epidemic is far from coming to an end by means of herd immunity. Early release of intervention intensity potentially increased a risk of the second outbreak.

## 1 Introduction

Throughout human history, Infectious diseases (ID), also known as transmissible diseases or communicable diseases, are considered as serious threats to global public health and economics [1]. From the 1918 influenza pandemic in Spain resulting in nearly 50 million deaths in 1920s, to recent ongoing global outbreaks of corona-virus disease 2019 (COVID-19) killing over 11 thousands people in all over the world [2], infectious disease is a leading contributor to significant mortality and causes huge losses to society as well as personal family burden. Among a variety of factors leading to emergence and outbreaks of ID, the key issues are population density and human mobility where in these cities with developed transportation systems, pathogens can be spread to large geographic space within a short period of time. For instance, the ongoing global epidemic outbreak of COVID-19 has spread to at least 146 countries and territories on 6 continents in 2 months. In order to give an accurate prediction of outbreaks, many researchers have been working in traditional ID propagation models [3–7] like SIR, SEIR, etc, for understanding COVID-19 transmission with human mobility and predicting outbreak process of epidemics. Meanwhile, as realizing a long period of this battle against COVID-19, many of them recently focus on intervention strategies [8–10] that can balance a trade-off between limited human mobility and potential economic loss in COVID-19 control. It poses an important research area that explores how and when to take what level of interventions in light of multiple natures and capabilities of countries.

In traditional compartmental models paradigm in epidemiology, SIR (Susceptible-Infectious-Recovered) [3] and SEIR (Susceptible–Exposure-Infectious-Recovered) [4] are two popular approaches to simulate and predict how infectious disease is transmitted from human to human. These two models have defined several variables that represent the number of people in each compartment at a particular time. As implied by the variable function of time, these models are dynamic to reflect the changes and fluctuations of these numbers in each compartment over time. For COVID-19 control in Wuhan, Zhong, et.al [11] introduced a modified SEIR model in prediction of the epidemics trend of COVID-19 in China, where the results showed that under strong suppression of "lockdown Hubei", the epidemic of COVID-19 in China would achieve peak by late February and gradually decline by the end of April 2020. Some other extended models [8] [12] are also proposed for predicting the epidemics of COVID-19 in Wuhan and give some similar forecasts. While above methods demonstrate good performance in prediction of COVID-19 outbreak by taking strong public intervention, also named as suppression strategy [13] that aims to reverse epidemic growth, one important challenge is that taking suppression strategy only is to treat disease controls as single-objective optimization of reducing the overall infectious populations as soon as possible, and require strategic consistency in a long term. In real-world, taking public health intervention strategies is actually a multiple-objective optimization problem including economic loss and society impacts. Thus, most countries have taken different intervention strategies, like enhanced surveillance and isolation to affected individuals in Singapore [14], four-stage response plan of the UK [15–16], mitigation approaches [13] and even multiple interventions taken in many

EU countries [17–21]. Due to the fact that standalone intervention strategy has apparent merits and limitations, it becomes highly necessary to study the feasibility of intervention strategies to certain country in light of its multiple natures and capabilities.

Targeting at this problem, this paper conducts a feasibility study that analyses and compares mitigation and suppression intervention strategies for controlling COVID-19 outbreaks in Wuhan and London. As shown in Fig 1 with Wuhan as a simulated case using data from [6],

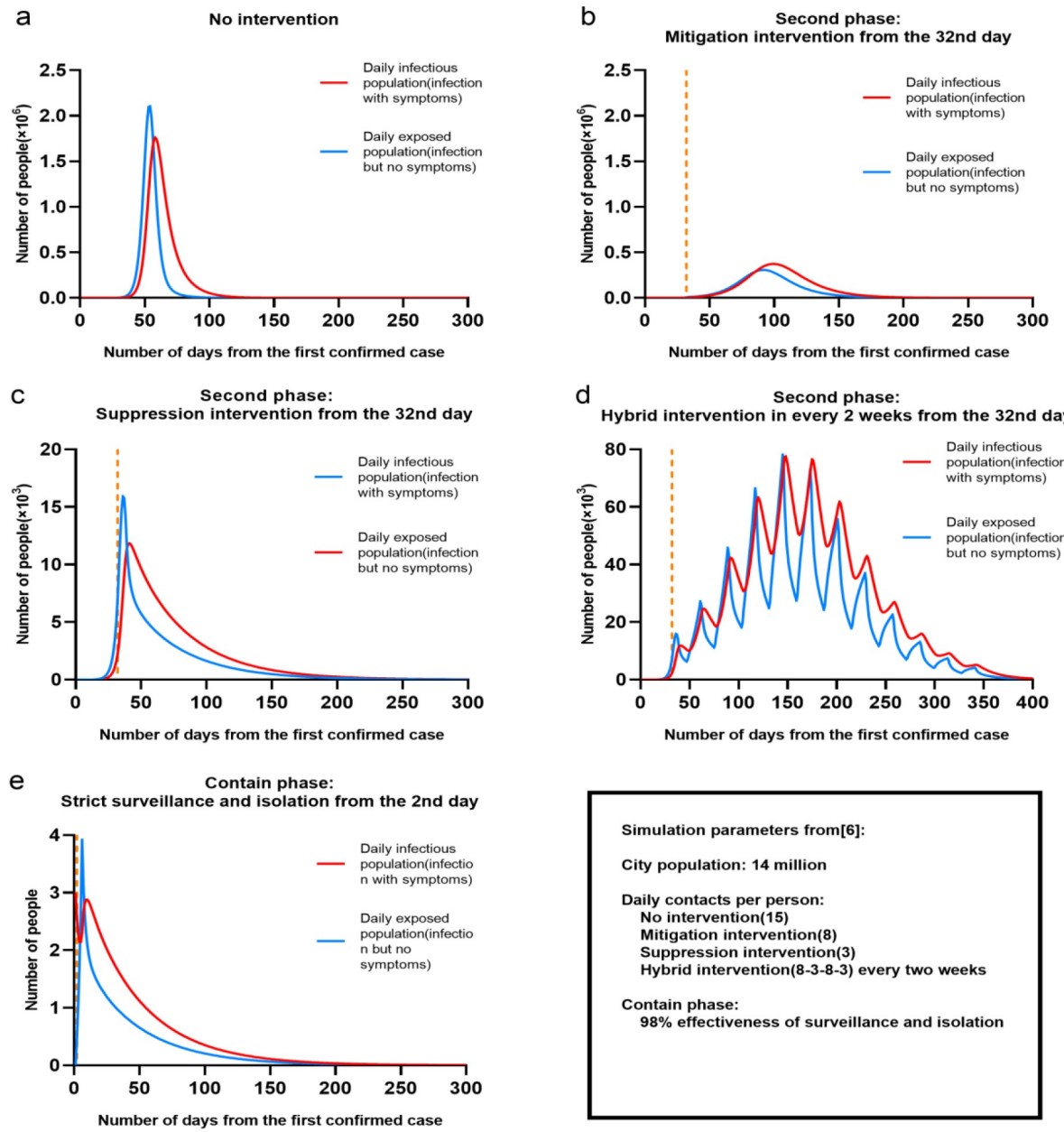

**Fig 1.** Illustration of controlling Wuhan COVID-19 outbreaks by taking different intervention strategies with parameters (City populations: 1.4 million; daily contacts per person: **(a)** No Interventions (15), **(b)**Mitigation Intervention(8), **(c)**Suppression Intervention (3), and **(d)**Hybrid intervention (8-3-8-3) every two weeks; **(e)**Contain phase (98% effectiveness of surveillance and isolation)). Note that a, b; c, d and e have different scales. The orange dotted line on the X axis represents the date of implementation of the intervention, b, c, d is the 32nd day. e is the 2nd day.

we demonstrated performance of taking different intervention strategies: a) No interventions: the peak of daily infections would be up to 2.1 million, but will be completed in 150 days. The epidemics lasts a relatively shorter period of 140–150 days, but lead to more death. b). Mitigation intervention from the 32nd day: the peak of daily infectious populations increased to 27.7 thousand, but the period of maintenance extended to 150 days. It implied there would be growing death, but less economic loss compared to suppression. c). Suppression intervention from the 32nd day: the peak of daily infections greatly reduced to 16 thousand, but it had to be followed at least 200 days. Nearly 3 months suppression may potentially lead to economic loss even crisis. d). Hybrid intervention of taking both suppression and intervention every 2 weeks: the epidemics of COVID-19 appeared a long-term multimodal trend where the peaks of daily infectious populations were within a range of 40–60 thousand. This might lead to less daily critical cases and offer more time to hospital for releasing their resources. e) Contain phase: taking 90% effectiveness of surveillance and isolation from the 2nd day of confirmed case potentially enables controlling a new outbreak of COVID-19, but it needs to be maintained over 300 days.

Above analysis demonstrates the complexity of controlling COVID-19 outbreaks that how and when to take what level of interventions. In this paper, we proposed a mathematical model: SEMCR to study this problem. The model extended traditional SEIR (Susceptible-Exposed-Infectious-Recovered) model [3, 4] by adding one important fact: there has been a direct link between Exposed and Recovered population. We found that the number of confirmed diagnoses reported by governments of various countries is actually much smaller than the actual number of infected persons. Although many exposed people have been infected, they will not show symptoms and transformed into infected people after the incubation period of the virus, and directly become recovered people after a certain self-healing cycle [22–23]. Because they are asymptomatic, they will not take the initiative to go to the hospital for treatment testing, so they will not be counted as the actual number of infected people. If we make a statistic on the cumulative number of cured people and the cumulative number of infected people, the difference between them is the number of asymptomatic patients who are directly transformed into cured people without going through the infection stage. Then, it defined parameters to classify two stages of COVID control: active contain by isolation of cases and contacts, passive contain by suppression or mitigation. The model was fitted and evaluated with public dataset containing daily number of confirmed active cases including Wuhan, London, Hubei province and the UK during January, 2020 and March 2020. For each point, we design and set up experimental protocols for comparison and exploration, highlighting following contributions:

- Immediate suppression taken in Wuhan significantly reduced the total exposed and infectious populations, but it has to be consistently maintained at least 90 days (by the middle of April 2020). Its success heavily relied on sufficiently external support from other places of China. This mode was not suitable to other countries that have no sufficient resources.

- In London, it is possible to take a hybrid intervention of suppression and mitigation for every 2 or 3 weeks over a longer period to balance the total infections and economic loss. While the total infectious populations in this scenario would be possibly 2 times than the one taking suppression, economic loss and recovery of London would be less affected.

- Both in Wuhan and London, one important issue of fitting practical data was that there were a large portion (probably 62.9% in Wuhan) of self-recovered populations that were asymptomatic or mild symptomatic. These people might think they have been healthy at home and

did not go to hospital for COVID-19 tests. Early release of intervention intensity potentially increased a risk of the second outbreak.

- One limitation of our model was that our prediction of infections and deaths depended on a parameter estimation of intervention intensity that presented by average-number contacts with susceptible individuals as infectious individuals in a certain region. It assumed that each intervention had equivalent effects on the reproduction number R in different regions over time. The measures of strong intervention in different countries and regions are similar, so the culture or other issues of different countries will not change the impact of strong intervention on the basic regeneration number.

The remainder of this paper is arranged as follows. Section 2 introduces the model. In the Section 3, the materials and implementation of experiment are reported. Section 4 provides detailed experimental evaluation and discussion. The conclusion and future directions are given in Section 5.

## 2 Methodology

### 2.1 Problem formulation of COVID-19 outbreak

We modified a SEIR model to account for a dynamic Susceptible [S], Exposed [E], Infectious [I] and Recovered [R] or Dead [D] population's state by extending two components: Mild [M] cases and Critical [C] cases. The modified modal is shown in Fig 2. Here, we assume that susceptible population $E$ represents susceptible population of a certain region; and $\beta$ represents effectiveness of intervention (strict isolation) in contain phase. If effectiveness of intervention

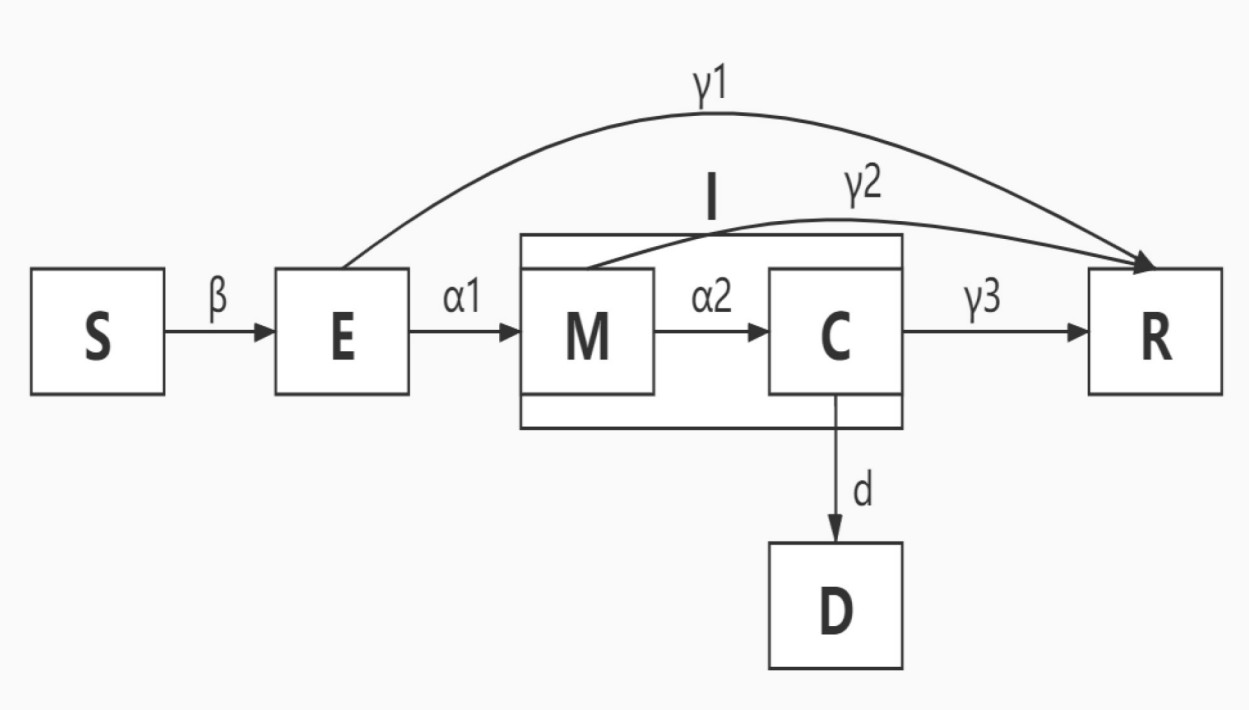

**Fig 2. Extended SEMCR model structure: The population is divided into the following six classes: Susceptible, exposed (and not yet symptomatic), infectious (symptomatic), mild (mild or moderate symptom), critical (severe symptom), death and recovered (ie, isolated, recovered, or otherwise non-infectious).**

in contain phase is not sufficiently strong, susceptible individuals may contract the disease with a given rate when in contact with a portion of exposed population (asymptomatic but infectious) $E$. After an incubation period, the exposed individuals become the infectious population $I$ (symptomatic) at a ratio $\alpha_1$. Notably, infectious population starts from mild cases $M$ to critical cases $C$ at a ratio $\alpha_2$. Finally, a portion $d$ of critical cases lead to deaths; the rest of infectious population will be recovered.

There are two enhanced features in our model in comparison to popular SEIR models [6, 8, 11]. The first one is a straightforward relationship between Exposed and Recovered population. We find that in the early outbreaks of COVID-19, some portion of exposed people may have no obvious symptoms or only develop as mild cases, but they cannot get a test due to lack of testing kits. This group of populations might be self-recovered in some days, but will not realize they were infected. And we can calculate the approximate proportion of the number of asymptomatic or mild symptoms in the total number of infected people by counting the cumulative number of each population. The second feature in our model is that we separate infectious population into mild and critical cases in light of their symptoms. According to the curve of the number of critical cases and the number of deaths, we can make a certain explanation for the relatively high mortality rate of early outbreaks of COVID-19 after Wuhan city took immediate inhibitory interventions on January 23, 2020.

Introducing above two features, it is helpful for evaluating real effects of different interventions. If we assumed the overall population of a certain region is $N$, the number of days is $t$, the dynamic transmissions of each components of our model are defined as follow:

$$\frac{dS(t)}{dt} = -\frac{\beta_1 S(t)I(t)}{N} - \frac{\beta_2 S(t)E(t)}{N} \tag{1}$$

$$\frac{dE(t)}{dt} = \frac{\beta_1 S(t)I(t)}{N} + \frac{\beta_2 S(t)E(t)}{N} - \alpha_1 E(t) - \gamma_1 E(t) \tag{2}$$

$$\frac{dI(t)}{dt} = \frac{dM(t)}{dt} + \frac{dC(t)}{dt} \tag{3}$$

$$\frac{dR(t)}{dt} = \gamma_1 E(t) + \gamma_2 M(t) + \gamma_3 C(t) \tag{4}$$

Regarding Mild cases, Critical cases and Death, the dynamic transmission is as below:

$$\frac{dM(t)}{dt} = \alpha_1 E(t) - \alpha_2 \frac{c+s}{m} M(t) - \gamma_2 M(t) \tag{5}$$

$$\frac{dC(t)}{dt} = \alpha_2 \frac{c+s}{m} M(t) - \gamma_3 C(t) - d\frac{c}{c+s} C(t) \tag{6}$$

$$\frac{dD(t)}{dt} = d\frac{c}{c+s} C(t) \tag{7}$$

Lastly, we define a benchmark in SEMCR model to reflect the strength of intervention over time, as $M_t$. It is presented by average number of contacts per person per day in a region.

## 2.2 Implementation of dynamic transmission of SEMCR

We estimated changes in COVID-19 transmissibility over time via the effective reproduction number($R_t$), which represents the mean number of secondary infections that result from a primary case of infection at time t. Values of $R_t$ exceeding 1 indicate that the epidemic will tend to grow, whereas values below 1 indicate that the epidemic will tend to decline. We estimated the time-varying reproduction numbers from serial intervals and incidence of COVID-19 cases over time.

In practical cases, it also needs to estimate the defined parameters including $\alpha_1$, $\alpha_2$, β, and $\gamma_1$, $\gamma_2$, $\gamma_3$, b, where β is the product of the people exposed to each day by confirmed infected people (k) and the probability of transmission (b) when exposed (i.e., β = $R_t$/γ = kb) and σ is the incubation rate which is the rate of latent individuals becoming symptomatic (average duration of incubation is $1/\alpha_1$). According to recent report [24], the incubation period of COVID-19 was reported to be between 1 to 14 days, we chose the midpoint of 6 days. Preliminary data suggests that the time period $\alpha_2$ from onset to the development of severe disease, including hypoxia, is 1 week [24]. γ is the average rate of recovery or death in infected populations. Using epidemic data from [11], we used SEMCR model to determine the probability of transmission (b) which was used to derive β and the probability of recovery or death (γ). The number of people who stay susceptible in each region was similar to that of its total resident population. Other transmission parameters were estimated with early prediction of Hubei cases in [11] on January 23 2020 using Monte Carlo simulation, as shown in the Table 1

Notably, as for the strength of intervention M, it was related to the population density in a region. We used a benchmark reported in [11] that assumes Hubei province with no intervention as M = 15, and after suppression intervention, M reduced to 3. When applying SEMCR model into other simulated cases, M was initialized according to the population density and human mobility in these places. Also, after taking any kind of interventions, the change of M would follow a reasonable decline or increase over few days, not immediately occur at the second day.

Following previous assumptions, the implementation of dynamic transmission of SEMCR model follows steps as below:

$$S_{t+1} = S_t - \frac{\beta_1 \times M_t \times I_t \times S_t}{N_t} - \frac{\beta_2 \times M_t \times E_t \times S_t}{N_t} \tag{8}$$

**Table 1. Parameters estimation in SEMCR model.**

| Name | Representation | Value |
|---|---|---|
| N | Total number of population in a region | N/A |
| $\beta_1$ | Transmission rate for the I to S | 0.157 |
| $\beta_2$ | Transmission rate for the E to S | 0.787 |
| $\alpha_1$ | Incubation period | 6 |
| $\alpha_2$ | Incubation period from M to C | 7 |
| $\gamma_1$ | Average period from E to R | 0.283 |
| $\gamma_2$ | Average period from M to R | 7 |
| $\gamma_3$ | Average period from C to R | 14 |
| d | Average period from C to D | 28 |
| m | Mild proportion | 0.8 |
| s | Severe proportion | 0.138 |
| c | Critical proportion | 0.061 |
| $M_t$ | Intervention intensity | 3–15 |

$$E_{t+1} = E_t + \frac{\beta_1 \times M_t \times I_t \times S_t}{N_t} + \frac{\beta_2 \times M_t \times E_t \times S_t}{N_t} - \alpha_1 E_t - \gamma_1 E_t \tag{9}$$

$$M_{t+1} = M_t + \alpha_1 E_t - \alpha_2 \times \frac{s+c}{m} \times M_t - \gamma_2 M_t \tag{10}$$

$$C_{t+1} = C_t + \alpha_2 \times \frac{s+c}{m} \times M_t - \gamma_3 C_t - d \times \frac{c}{s+c} \times C_t \tag{11}$$

$$I_{t+1} = M_{t+1} + C_{t+1} \tag{12}$$

$$D_{t+1} = D_t + d \times \frac{c}{s+c} \times C_t \tag{13}$$

$$R_{t+1} = R_t + \gamma_1 E_t + \gamma_2 M_t + \gamma_3 C_t \tag{14}$$

### 2.3 Model evaluation protocol

In order to utilize our proposed SEMCR model into practical cases, we design an evaluation protocol to access multiple effects of taking different intervention strategies to control outbreak of COVID-19 in 4 typical cases, including Hubei province, Wuhan city, the UK and London, as shown in Fig 3.

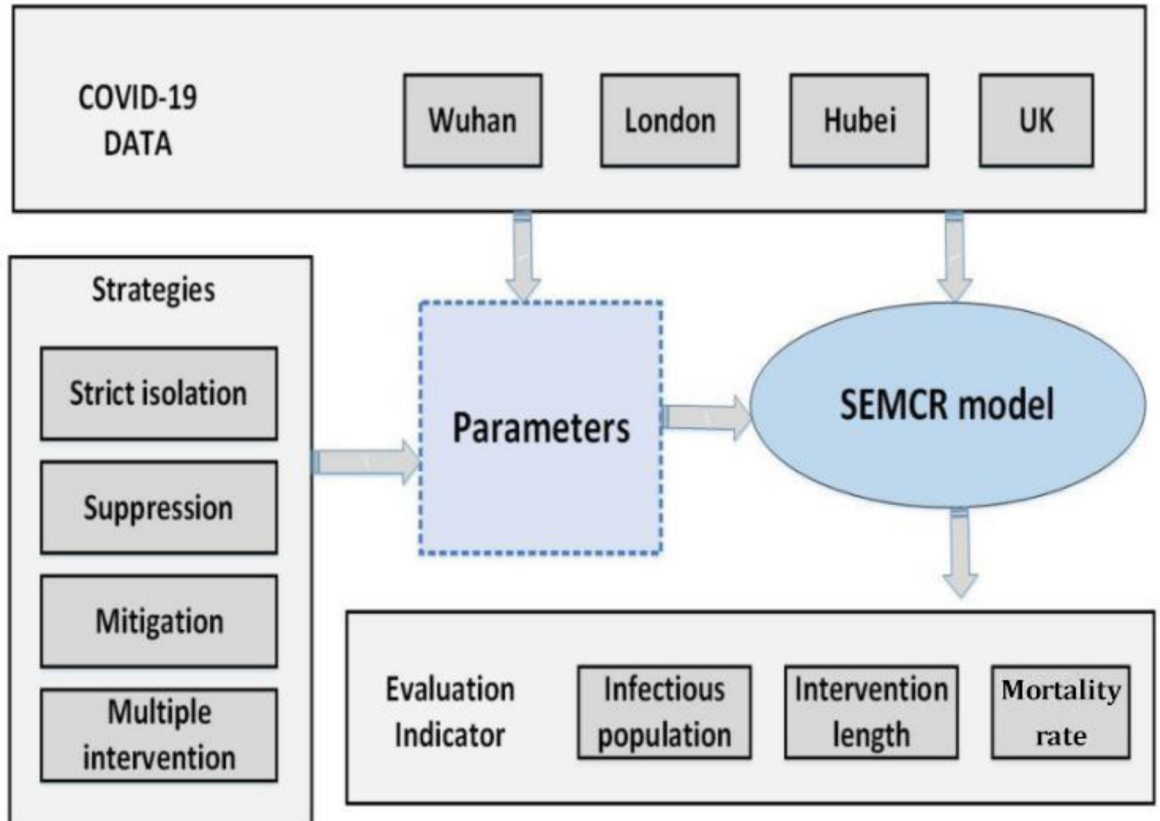

**Fig 3. SEMCR model evaluation protocol.**

The first stage is initial parameters estimation using COVID-19 data from four cases: Hubei, Wuhan, UK and London. In this stage, a preliminary qualitative assessment of each case is performed, by comparing their similarity and dissimilarity on area, transportation, population density, migration flows, date of the first confirmed case, etc. We would determine value of initial parameters in the SEMCR including N, $M_t$, and date of the first confirmed case. Notably, in Wuhan, the date of the first confirmed case is not officially released. The work [8] estimates the first confirmed case is by the end of Nov 2019; and Zhong [11] points out the first confirmed case in Wuhan is on 23nd December 2019. Here, we take the same settings of first confirmed case on 23nd December 2019.

The next step is to estimate and normalize other parameters in the model. Assuming that COVID-19 has similar transmission ratio and incubation rate in all four cases, we use parameter values fitted from [11], where incubation rate is 1/6; the rate of transmission for the I to S is 0.157; the rate of transmission for the E to S is 0.787. As for estimation of other parameters, we follow the COVID-19 official report from WHO [24], including the proportion of Mild, Severe and Critical cases, the probability of death, etc.

Thirdly, how to take intervention strategies needs to be evaluated by tuning parameters in SEMCR model. The key tuning operation is to adjust the level of $M_t$ over a period. For instance, we assume that no intervention strategies result in unaltered internal mobility of a region, taking suppression strategy in Wuhan means a reduction of M to 3. But in other cases with larger area, it is extremely difficult to take a complete suppression strategy. So the reduction of M will be relatively adjusted to 4 or 5.

Final stage, we perform quantitative analysis of effectiveness of different intervention strategies, including: strict surveillance and isolation, suppression strategy, mitigation strategy, and multiple intervention. The evaluation metric of cross-validation is employed to evaluate the performance of COVID-19 progression model. The final two evaluation indicators are the length of intervention and the peak time. The length of intervention is calculated due the date that confirmed cases are nearly clear to zero.

## 2.4 Data collection

The most recent epidemiological data in Hubei Province, China based on daily COVID-19 outbreak numbers reported by the National Health Commission of China were retrieved [25]. The dataset used to analyse several European countries comes from the statistics of worldmeters [26]. Specifically, we used confirmed cases, new cases, recovery cases and deaths since 22nd January 2020 to 25th March 2020. All the datasets used in this paper are anonymised. Also regarding the daily update from world meter, we record the number of confirmed cases and death each day in four cases. In order to simulate four cases, we require the exact confirmed cases in the first 2–5 days to initialise parameters of our model.

## 3 Experiments

### 3.1 Evaluation of cases with no interventions

We simulated four cases that predict COVID-19 outbreaks without taking any interventions. The initial populations were given as London (9.3 million), Wuhan (14.18 million), UK (66.49 million) and Hubei (58.9 million). The parameter M representing average number of contacts per person per day was given as 15 to London, Wuhan and Hubei; 12 to the UK. The simulation results were given in Fig 4.

The results showed that in the peak time, there would be up to 1.16 million, 2.1 million, 4.5 million and 8.7 million Exposed population (infection but no symptoms) at London, Wuhan, UK and Hubei. This implied that: 1) the total infectious population of these four cases would

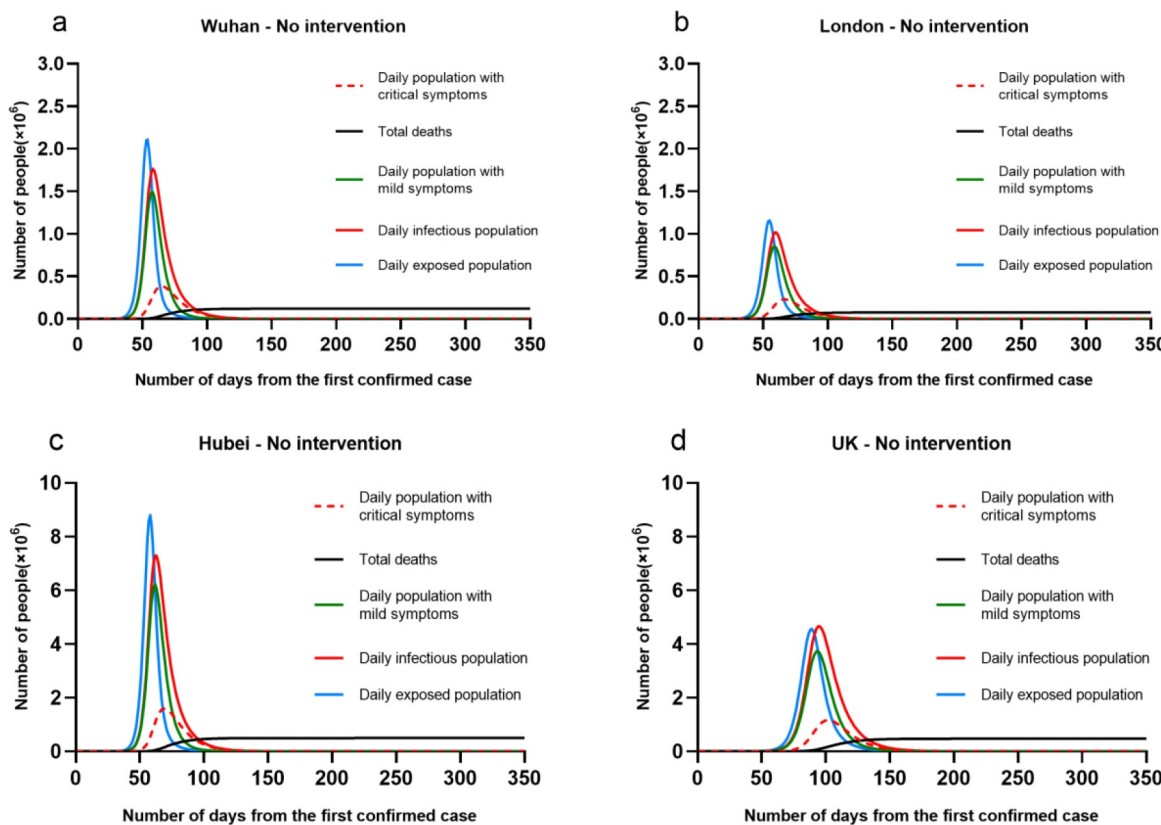

**Fig 4. Four cases (Wuhan, London, Hubei and UK) with no intervention.**

be 7.76 million in London, 12.27 million in Wuhan, 50.95 million in Hubei and 48.46 million in the UK. 2) The total death of these four cases would be 76 thousand in London, 120 thousand in Wuhan, 493 thousand in Hubei and 475 thousand in the UK. It equalled to about over 80% of total population of each region will be infectious, with the mortality rate nearly 1%. It showed that without intervention, the outbreak of COVID-19 would lead to huge infections and deaths. The main reason was that COVID-19 was estimated as relatively high production number $R_0$ up to 2–2.5 [24], where the transmission ratio $\beta_2$ from Susceptible to Exposed is up to 78%. Thus, in some regions with high migration and dense population, it would easily lead to an outbreak.

But, one notable issue was that the initialization of parameter M seems to impact on the occurrence of peak time and the length of overall period. In the cases of London, Wuhan and Hubei with M = 15, their peak date were all roughly on the 60[th] day from the date of first confirmed case; but UK with M = 12, their peak time was delayed to the 100[th] day from the date of first confirmed case. That meant to regions with similar total population, low population density potentially reduced overall infectious population, but delayed the peak time of outbreak as a result of longer period.

## 3.2 Effectiveness of surveillance and isolation

As for the strategy of taking surveillance and isolation intervention in the contain phase, recent study [15] developed a stochastic transmission model parameterised to the COVID-19 outbreak. It proved that highly effective contact tracing and case isolation can control a new

outbreak of COVID-19 within 3 months, where for a production number $R_0$ of 2–2.5, more than 90% contacts had to be traced. We transferred this finding as a tuning parameter $\beta_2$ to evaluate if the outbreak of COVID-19 could be controlled.

Considering that 98% of contacts had to be traced, it implied that the surveillance and isolation was effective to scale down the group of contacts. In other words, we simulated a situation that from the second day of receiving confirmed case, only 5% of the overall population would be possible to contact the infectious ones. Then, the $\beta_2 = 0.05$. The results were shown in the Fig 5. The results show that in all four cases, the outbreaks of COVID-19 were successfully controlled, and had no peak time. There would be only more than 40 people that were infection and finally recovered at London, Wuhan, UK and Hubei. The overall period of COVID outbreak was less than 100 days. This finding was as similar as the proof in [15].

Notably, the simulated situation of Wuhan, London and Hubei were same, it was mainly because we initialised the same value of confirmed cases as 3, and M = 15 in all cases. When taking highly effective surveillance and isolation, the transmission of COVID-19 was limited and controlled within a small group of population. The population difference would not affect the total infectious ones. But in the UK case with M = 12, low population density limited the effectiveness of surveillance and isolation, as a result of more infectious population. It implied another fact that to these countries with low population density, it was challenging to take high quality of intervention like surveillance and isolation.

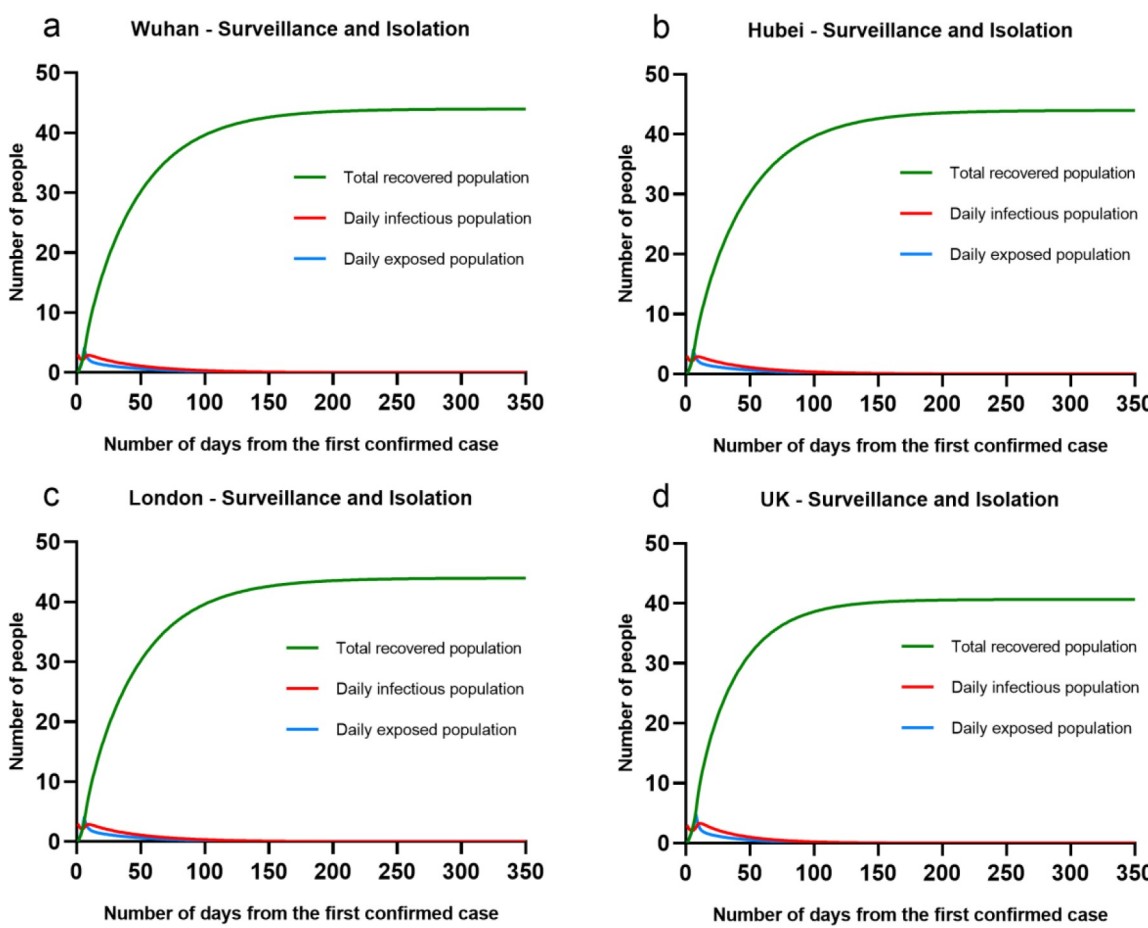

**Fig 5. Four cases (Wuhan, London, Hubei and UK) by taking isolation in contain phase.**

Then, we evaluated when the outbreak of COVID-19 could occur by adjusting the value of parameter $\beta_2$ in the UK case. As shown in Table 2, we recorded the total recovered population, and if there would be a peak of outbreak. As the increased $\beta_2$, the total recovered population was dramatically increased and generated a peak as $\beta_2 = 0.075$. That meant that if we cannot guarantee at least 92.5% of UK population being not contacted by infectious people, it would be indispensable to have an outbreak of COVID-19. If the isolation intervention was less effective; the more population would be infectious; and the peak day was brought forwarded. In the early stage, taking high effective surveillance and isolation in any regions is necessary to avoid the outbreak of COVID-19.

## 3.3 Effectiveness of suppression intervention

The suppression strategy was recognized as the most effective solution to reduce the infectious population, where it was taken in Wuhan on 23rd January 2020. In the work [11], Zhong has reported that taking suppression strategy has successfully limits the overall infectious population in Hubei on 22nd February 2020 to 50K. And if the suppression strategy was taken one week earlier, this figure would reduce to 18K. Thus, we would like to simulate the situation of taking suppression strategy in London. The parameter M was given as 15 to 4 in London, in comparison to Wuhan from 15 to 3. The simulation results were given in Fig 6.

After taking intensive suppression on 23rd March in the UK, the change trend of the basic regeneration number in the UK and Wuhan is consistent. A rapid decline in R has occurred in later March, from 2.81[1.16–5.19] at the 24th day (1st March 2020) to 0.68[0.58–0.79] at the 51st day (28th March 2020). It implied implementing suppression in the UK performed significantly impact on reduction of infections.

The results showed that taking suppression strategy, the cases in Wuhan and London appeared a similar trend that daily exposed and infectious population would be greatly reduced. The outbreak of COVID-19 was controlled by the 100th days, and can be nearly ended by the 150th days. The difference was that the daily infectious population of London was nearly double to the ones in Wuhan. It was probably because we simulated the date of taking suppression strategy in Wuhan (the 32nd day) is 3 days earlier than London (the 35th day). Another possible reason was that considering the impact of culture difference, the value of M was only limited to 4 in London, but not as lower as 3 in Wuhan. In fact, the suppression in Wuhan was actually applied to limit mobility in community level with very high intensity, which was hard to be followed by London.

We considered another two situations of taking suppression intervention 1 week earlier or 1 week later in London, as shown in Fig 6. The results showed that the overall infectious population would be greatly reduced to 3.5 thousand if taking actions one week earlier; oppositely it would increase to 178 thousand infection if taking actions one week later.

**Table 2. Parameter tuning in the UK case.**

| Value | Total recovered population | Peak | Peak Day |
|---|---|---|---|
| 0.055 | 190 | N | N |
| 0.060 | 1100 | N | N |
| 0.065 | 2742 | N | N |
| 0.070 | 130K | N | N |
| 0.075 | 1.90M | Y | 240 |
| 0.080 | 3.50M | Y | 230 |
| 0.090 | 4M | Y | 180 |

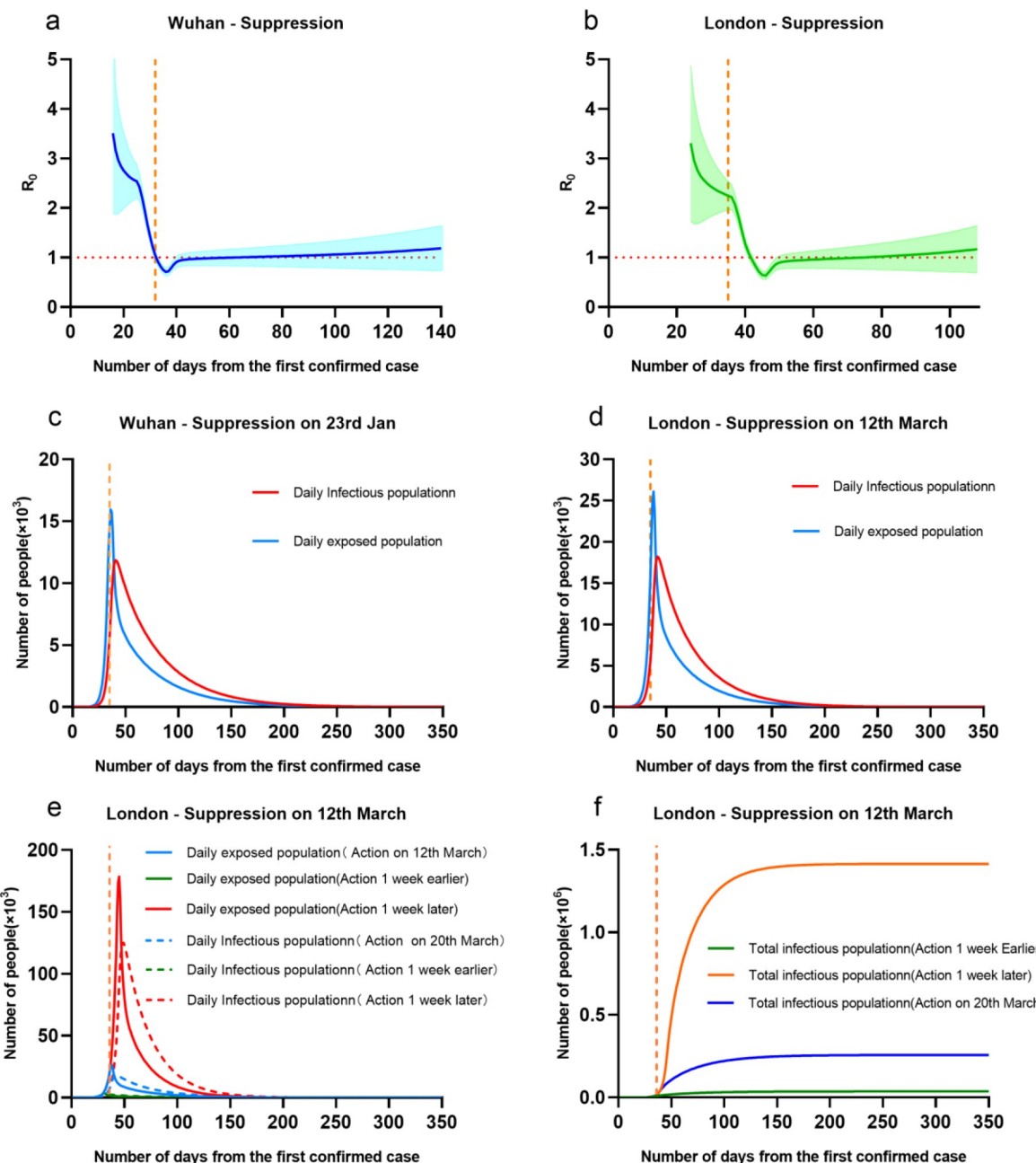

**Fig 6.** **(a, c)** Wuhan and **(b, d, e, f)** London by taking suppression intervention. Note that a,b; c,d,e and f have different scales. And the orange dotted line on the X axis represents the date of implementation of the intervention, a, c is the 32nd day, b, d, e, f is the 35th day.

We used our model to estimate the impacts of suppression on controlling infections of other 6 EU countries (Italy,Spain, Germany, France, Belgium and Switzerland), as shown in Table 3. Most suppression in other countries began around $10^{th}$ -$17^{th}$ March (the $28^{th}$– $40^{th}$ day from first confirmed case).For each country, we model the number of infections, the number of deaths, and R, the effective reproduction number over time. Specific interventions are assumed to have the same relative impact on R in each country when they were introduced there and are informed by mortality data across all countries.

**Table 3. Forecasts for six other European countries.**

| Countries | Total infectious population | Total dead population | Mortality rate |
|---|---|---|---|
| Italy | 1.9 million | 41 thousand | 2.17% |
| Spain | 2 million | 41 thousand | 2.07% |
| Germany | 0.77 million | 11 thousand | 1.5% |
| France | 1.6 million | 33 thousand | 2.09% |
| Belgium | 0.62 million | 12 thousand | 1.93% |
| Switzerland | 0.23 million | 3.4 thousand | 1.47% |

As shown in Table 3, in Italy from 8th February 2020, the total number of infections is about 1.9 million, the total number of deaths is about 41 thousand, the true mortality rate is 2.17%. In Spain from 1st February 2020, the total number of infections is 2 million, the number of deaths is 41 thousand, the true mortality rate is 2.07%. In Germany from 11th February, the total number of infections is about 0.77 million, and the number of deaths is about 11 thousand. The true mortality rate is 1.5%. In France from 15th February 2020, the total number of infections is about 1.6 million, and the number of deaths is about 33 thousand. The true mortality rate is 2.09%. In Belgium from 15th February 2020, the total number of infections is about 0.62 million and the number of deaths is about 12 thousand. The true mortality rate is 1.93%, and. lastly, in Switzerland 19th from February 2020, the total number of infections is 0.23 million people, the number of deaths is about 3.4 thousand people, the real mortality rate is 1.47%.

The time of intervention in these six European countries is similar, but from the analysis of the infections in these six European countries can be seen that the intervention effects of Germany, Belgium and Switzerland are better, and the mortality rate of Germany and Switzerland is less than Four other countries. From the data we collected can be seen that Germany and Switzerland have adequate medical resources, while Italy, Spain, France and Belgium have insufficient medical resources. This corresponds to the difference between the number of deaths and mortality we predicted. The above also confirms the previously mentioned point of view. When medical resources are insufficient, premature intervention may cause strain on medical resources, leading to more deaths and higher mortality.

Another two important issues of taking suppression strategy were intervention length and the peak time. In order to effectively control the outbreak, the intervention length of taking actions at above three timing points all required at least 100 days. It potentially led to significant side effects to economic and society, including job loss, mental health, etc.Also, the peak time arrived earlier than its nature transmission if taking actions one week earlier. This might increase a shorter time to government for preparing sufficient resources, and causing more difficulties in control the outbreak of COVID-19.

## 3.4 Effectiveness of mitigation intervention

The mitigation strategy was recently highlighted by researchers from Imperial College [27, 28], where this strategy was initially taken by UK government. The aim of mitigation was to use other strategies to help individuals so that not to interrupt transmission completely, but to reduce the health impacts of an epidemic. In this cases, population immunity built up through the epidemic, leading to an eventual rapid decline in case numbers and transmission dropping to low levels. Thus, we simulated the situation of the UK by taking mitigation strategy with different level of strengths. The parameter M representing average number of contacts per person per day was given as 12 to the UK, and gradually reduced to 10 and 8 from the 32[nd] day of first confirmed case. The simulation results were given in Fig 7A and 7C.

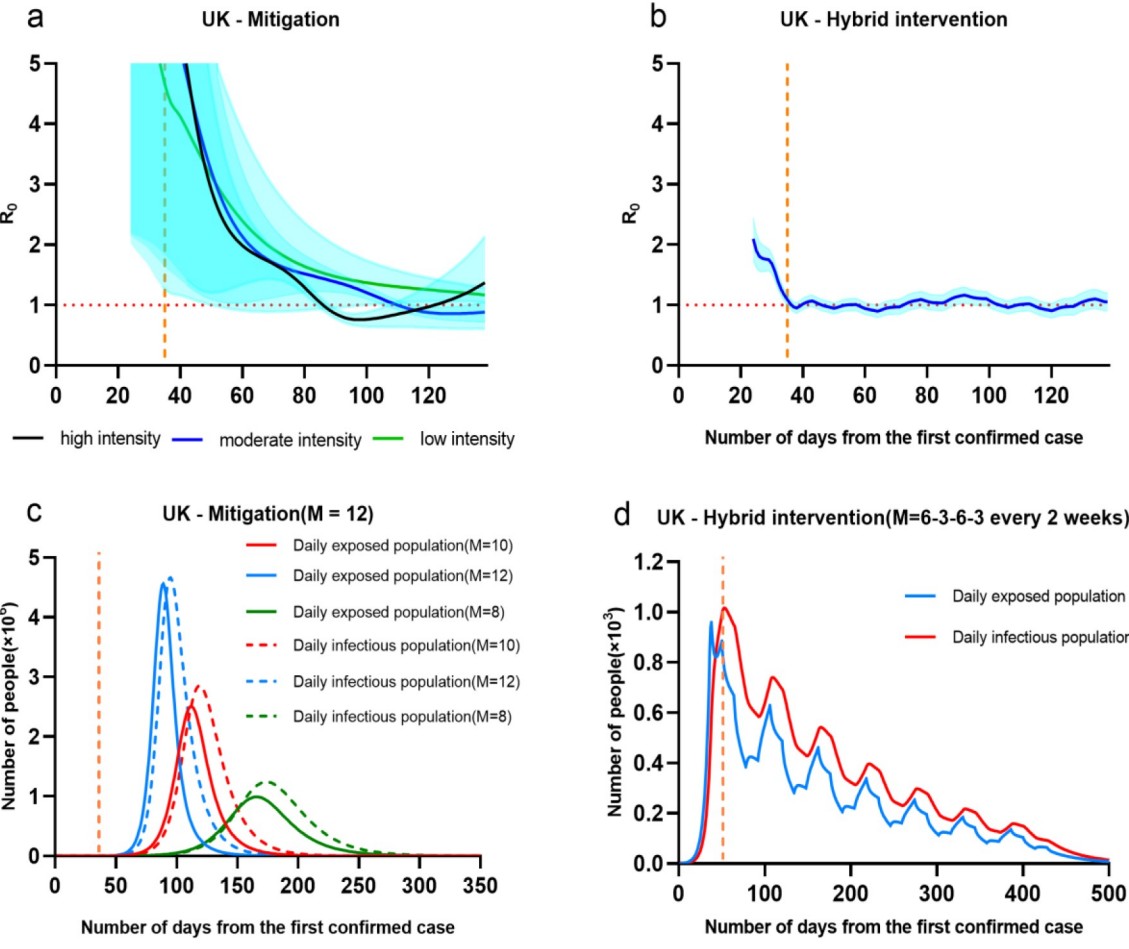

**Fig 7. (a, c)** UK by taking Mitigation or **(b, d)** Hybrid intervention. Note that a, b; c and d have different scales. And the orange dotted line on the X axis represents the date of implementation of the intervention, a, b, c, d is the 35th day.

The simulation of basic regeneration numbers(R) compared to suppression strategy, mitigation strategy taken in the UK gave a slower decline in R in March, from 2.73[0.97–5.40] on the 24th day (1st March 2020) to 0.98[95% CI 0.88–1.09] on the 110th day (27th May 2020). It implied that before R drops below 1, there were still much growth of infections in the UK. And it can be seen from the Fig 7 that the more relaxed the intervention, the longer it takes R to fall below 1.

The results showed that taking mitigation strategy was effective to reduce daily infectious population, further lead to huge reduction of total infectious population. By reducing the parameter of M from 12 to 10 or 8, the peak of daily infectious population reduced from 4.5 million to 2.5 million or 1 million. On the other hand, the peak time of infectious population would be delayed as taking mitigation intervention, where the peak dates of infectious population at M = 12, 10 and 8 were roughly on the 80th, 110th and 170th days. The overall period of outbreak would be extended from 180 days to 200 or 250 days. Above simulated figures appeared a similar trend as findings from [25]. Taking mitigation intervention in the UK was capable of reducing the impact of an epidemic by flattening the curve, reducing peak incidence and overall death. While total infectious population may increase over a longer period, the final mortality ratio may be minimized at the end. But as similar as taking suppression strategy,

the mitigation interventions need to remain in place for as much of the epidemic period as possible. However, the timing of introducing this mitigation intervention was important, where too early execution may allow transmission to return once they were lifted and sufficient "herd immunity" has not been developed.

## 3.5 Effectiveness of hybrid intervention

In terms of above discussion, the effectiveness of taking any one intervention (either suppression or mitigation) is likely to be limited. It is highly necessary to consider the possibility of taking multiple interventions to be combined to have a substantial impact on social and economic cost reduction. We simulated one simple situation of taking multiple strategy in London from the 35th day, by giving a hybrid of suppression and mitigation strategies every 2 weeks. The M was given as a pattern of 6–4.5-3-4.5–6 in London, where it meant mitigation and suppression strategies were taken in an every 2 weeks roll. For minimizing side effects of taking interventions on human mobility, the application of first mitigation to reduce M from 12 to 6 spends 2 days. The simulation results were given in Fig 7B and 7D.

The simulation of basic regeneration numbers(R) under the hybrid intervention in the UK shows that a similar early trend of R as suppression, where there was a fast decline in R in March, from 1.98[1.57–2.43] on the 24th day (1st March 2020) to 0.86[0.73–0.99] on the 51th day (28th March 2020). It implied that 3 weeks rolling intervention (M = 3 or 6) had equivalent effects on controlling transmissions as suppression, but need to be maintained in a longer period of 300 days. From then, R value was oscillated between 1.12 [0.95–1.29] and 0.86[0.71–1.02] with the shrinkage of intervention intensity.

The results showed that the epidemic appeared a multi-modal decline trend over 500 days. The first peak of infectious population occurred on the 53rd days with 1017 infections after taking suppression intervention to reduce M from 12 to 3. After two weeks, mitigation strategies were taken so that the second peak of infectious population raised up to 750 infections. The total infectious population was 53075 over 500 days; the deaths was limited to 520.

Apparently, taking multiple intervention in the UK is capable of reducing the impact of an epidemic by fluctuating the curve, reducing overall infections and death. While the total period will be extended, final mortality ratio may be minimised at the end. But the longer period of limiting human mobility might increase economic risks and reduce employment ratio. There will be plenty of choices to taking multiple interventions through adjusting the strength and length of intervention. The consequence possibility show a similar multimodal curve but with different peak incidence.

## 4 Discussion and limitation

As we mentioned in section 1, the question on how and when to take what level of interventions to control an epidemic is highly challenging, particularly in light of multiple natures and capabilities of countries. In many cases, it is even hard to evaluate effectiveness until the end of epidemic, and there is always controversy on taking any one intervention. However, our findings contribute to several useful suggestions on controlling COVID-19 outbreak.

The first point is that highly effective surveillance and isolation strategy is necessary to control an epidemic in early stage. Ideally, if this strategy can be executed in excellent level, there will be no huge outbreak later on. Considering the area, transportation, migration flows and population density of a region, most countries cannot achieve excellent level of isolation in contain phase, probably only in fine or good level. The outbreak of an epidemic is inavoidable. But considering its low social and economic cost, this strategy is still a cost-effective option.

Secondly, the cases in Wuhan and London approve high effectiveness of suppression strategy to reduce the overall infection. Probably in the UK or similar countries, suppression will minimally require a combination of social distancing of the entire population, home isolation of cases and household quarantine of their family members. But its practical effectiveness is not possible to achieve as same as Wuhan, as the success of suppression strategy in Wuhan is based on locking down human mobility to community level and sufficient resource support from other cities or provinces in China. If there are no sufficient external support, it will be risky to take intensive suppression to entire country due to huge impacts on its economics. Also, taking such intensive intervention to control an epidemic will need to be maintained until vaccine released (up to 12 months or more). If intensive interventions are relaxed at any time points, the transmission will quickly rebound. This is more like a multi-modal curve when taking multi-intervention strategies.

Thirdly, we also find out that while COVID-19 is estimated as a high production rate (R = 2–2.5) [24], experimental evaluation results show that in either Wuhan or London cases fitted with real data in the last 6 weeks, high percentage of exposed or infectious population (at least 62.9% of infectious population in Wuhan) are actually self-recovered. These people may have no or mild symptoms but been not checked as confirmed cases. This is one important issue that Zhong's SERI model [11] has ignored. It will answer the model [8] predicts practical infectious population in Wuhan that ten times over figures in [11]. Similarly, it could explain the estimation of practical mortality ratio can be varied in [11] and [8].

Lastly, one limitation of our model is that its prediction of infections and deaths depends on a parameter estimation of intervention intensity that presented by average-number contacts with susceptible individuals as infectious individuals in a certain region. We assume that each intervention has the same effect on the reproduction number in different regions over time. The measures of suppression intervention in different countries and regions are similar, so the culture or other issues of different countries will not change the impact of suppression intervention on the basic regeneration number. As implementing hybrid intervention, the policy needs to be specific and well-estimated at each day according to the number of confirmed cases, deaths, mortality ratio, health resources, etc.

## 5 Conclusion

This paper conducts a feasibility study by defining a mathematical model named SEMCR that analyses and compares mitigation and suppression intervention strategies for controlling COVID-19 outbreaks in London and Wuhan Cases. The model not only fits and evaluates through public data sets containing the number of daily confirmed active cases in Wuhan, London, Hubei Province and the United Kingdom, but also uses the trained model to predict and analyze six other European countries at the end. The experimental findings show that the optimal timing of interventions differs between suppression and mitigation strategies, as well as depending on the definition of optimal. In the future, we can expand our model to realize the comparative analysis of the number of severe cases and the number of medical resources by adding medical resources, and optimize the timing and intensity of interventions to reduce the demand for medical resources.

## Author Contributions

**Conceptualization:** Po Yang, Gaoshan Bi.

**Data curation:** Jun Qi, Gaoshan Bi, Geng Yang.

**Funding acquisition:** Yun Yang, Bin Sheng.

**Investigation:** Po Yang, Yun Yang.

**Methodology:** Po Yang, Gaoshan Bi.

**Project administration:** Yun Yang, Bin Sheng.

**Resources:** Jun Qi, Bin Sheng.

**Software:** Jun Qi, Shuhao Zhang, Xulong Wang.

**Supervision:** Geng Yang.

**Validation:** Shuhao Zhang, Xulong Wang.

**Writing – original draft:** Po Yang.

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
