## [Decision Letter · Decision Letter 0]

27 May 2020

PONE-D-20-10226

Feasibility Study of Mitigation and Suppression Intervention Strategies for Controlling COVID-19 Outbreaks in London and Wuhan

PLOS ONE

Dear Dr. Yang,

Thank you for submitting your manuscript to PLOS ONE. After careful consideration, we feel that it has merit but does not fully meet PLOS ONE’s publication criteria as it currently stands. Therefore, we invite you to submit a revised version of the manuscript that addresses the points raised during the review process.

We look forward to receiving your revised manuscript.

Kind regards,

Abdallah M. Samy, PhD

Academic Editor

PLOS ONE

2. In ethics statement in the manuscript and in the online submission form, please provide additional information about the database used in your retrospective study. Specifically, please ensure that you have discussed whether all data were fully anonymized before you accessed them and/or whether the IRB or ethics committee waived the requirement for informed consent. If patients provided informed written consent to have their data used in research, please include this information.

3. Please respond by return e-mail with an updated version of your manuscript to include your abstract after the title page.

4. Please upload a copy of Figure 8, to which you refer in your text on page 15. If the figure is no longer to be included as part of the submission please remove all reference to it within the text.

5. We suggest you thoroughly copyedit your manuscript for language usage, spelling, and grammar. If you do not know anyone who can help you do this, you may wish to consider employing a professional scientific editing service.  

(Note: HTML markup is below. Please do not edit)

**Reviewers' comments:**

Reviewer's Responses to Questions

**Comments to the Author**

1. Is the manuscript technically sound, and do the data support the conclusions?

Reviewer #1: Partly

Reviewer #2: No

Reviewer #3: Partly

2. Has the statistical analysis been performed appropriately and rigorously? 

Reviewer #1: No

Reviewer #2: N/A

Reviewer #3: Yes

3. Have the authors made all data underlying the findings in their manuscript fully available?

Reviewer #1: Yes

Reviewer #2: Yes

Reviewer #3: No

4. Is the manuscript presented in an intelligible fashion and written in standard English?

Reviewer #1: No

Reviewer #2: Yes

Reviewer #3: Yes

5. Review Comments to the Author

Reviewer #1: The paper addresses an interesting and important approach to dealing with lockdown exit in a covid19 epidemic and the idea deserves to have a wider audience in academic and policy circles.

However, the paper is NOT novel in this, either in the specific ideas proposed for covid19 or the simulations. Multiple papers have been published prior to this paper (also prior to the pre-print released mid April), and others on the broader ideas of switching quarantines in disease outbreaks.

e.g. the first one directly on this topic:

On Fast Multi-Shot COVID-19 Interventions for Post Lock-Down Mitigation

M. Bin, P. Cheung, E. Crisostomi, P. Ferraro, H. Lhachemi, R. Murray-Smith, C. Myant, T. Parisini, R. Shorten, S. Stein, L. Stone, https://arxiv.org/abs/2003.09930 appeard 22nd March 2020.

a follow on paper published on April 7th

Containing COVID-19 with a two-day workweek, Uri Alon,

https://medium.com/@urialonw/adaptive-cyclic-exit-strategies-from-lockdown-to-suppress-covid-19-and-allow-economic-activity-4900a86b37c7

and expanded to :

https://www.medrxiv.org/content/10.1101/2020.04.04.20053579v4

also, related from April 3rd,

Alternating quarantine for sustainable mitigation of COVID-19

Dror Meidan, Nava Schulamann, Reuven Cohen, Simcha Haber, Eyal Yaniv, Ronit Sarid, Baruch Barzel

https://arxiv.org/abs/2004.01453

and the broader literature on switching quarantine:

X., Liu and P., Stechlinski, The Switched SIR Model. Infectious Disease Modeling, Nonlinear Systems and Complexity, vol 19. Springer, Cham, 2017.

L., Zhu, and Y., Zhou, The Dynamics of an SIQS Epidemic Model with Pulse Quarantine. IEEE Chinese Control and Decision Conference, pp. 3546-3551, 2008.

J., Lai, S., Gao, Y., Liu, and X., Meng, Impulsive Switching Epidemic Model with Benign Worm Defense and Quarantine Strategy. Complexity, 2020.

Liu, X., and Stechlinkski, P., Infectious Disease Modeling - A Hybrid System Approach, Nonlinear Systems and Complexity, Series Editor: Albert C.J. Luo, Springer Nature, 2017.

The authors propose a 'novel' addition in Section 2 where people can move from Exposed directly to Recovered if they have no symptoms, but there is no evidence provided of examples in the published literature that these people do not move through an infectious stage before being Recovered, which would invalidate some of the model findings. The idea of asymptomatic infectious is better modelled in

Giordano, G., Blanchini, F., Bruno, R., Colaneri, P., Di Filippo, A., Di Matteo, A., Colaneri, M., and the COVID19 IRCCS San Matteo Pavia Task Force, A SIDARTHE Model of COVID-19 Epidemic in Italy, 2020,

https://arxiv.org/abs/2003.09861. (since published in nature medicine)

which was the basis for the simulations of the Bin et al switching lockdown above.

The paper has very frequent errors in the English, in the text and figures. Too many examples to enumerate in the review.

There is an apparent error in equation (3), should it not be I(t) = M(t)+C(t) rather than dI(t)/dt as I is just a container for M and C? or dI(t)/dt = dM(t)/dt + dC(t)/dt?

There is an apparent error in equation (12) should it not be I_t = M_t+C_t rather than I_t+1 = M_t+C_t

So I think before being publishable the authors need to fix some of the model description errors, better justify the assumptions around asymptomatics and relate their approach to prior publications (long term and recent) in the field. Also they really need to clarify what extra benefit we get from their simulations compared to the previously published ones which often used more advanced models to test their simulations, and had more rigorous sensitivity analysis.

Reviewer #2: The paper title “Feasibility Study of Mitigation and Suppression Intervention Strategies for Controlling COVID-19 Outbreaks in London and Wuhan” modifies SERI model. Their SEMCR that access the effectiveness of mitigation, suppression and hybrid interventions for controlling COVID-19 outbreaks in London and Wuhan.

The work is very straight forward and its hard to see any significance. Comparing London with Wuhan seems not a good fit. Both cities have different culture and life style. Similarly, Wuhan was the first city hit by corona and many things were not known by then. London had ample time with more information (spread rate, incubation period) to deal with cOVID-19 disease.

More comments:

In the model, it is assumed that go R directly. It seems in covid-19 this assumption is not invalid. Even if someone gets recovered, he/she gets infected. May be the justification could be that no symptoms are shown and one recovered. However, still the person can infect others without showing symptoms. This point need discussion or any references from literature.

"We find that in the early outbreaks of COVID-19, some portion of exposed people may have no obvious symptoms or only develop as mild cases, but they cannot get a test due to lack of testing kits. This group of populations might be self-recovered in some days, but will not realize they were infected". The same is true even today, You cam consider undiagnosed cases too.

In many figures, 10 ^ is used. Proper units such as million for x-axis should be used in figures i.e, Thousand, million etc. 10 powers are hard to understand.

The statement “like immediate lock-down cities at epicenter or mitigation that slows down but not stopping epidemic for reducing peak healthcare demand’ needs explanation. This reviewer understands when you delay the spread, peak can be automatically lowered.

Abstract is too long, around 450 words.

Take R0 of U, and then London.

For clear understanding, you can compare figures for London “ with intervention” and "without intervention “ side by side.

Reviewer #3: In this work impact of strategies such as strict isolation, mitigation, and interventions are considered for COVID-19 spread in Wuhan and London. The paper is simple and discusses mainly a comparison of two cities based on limited data for COVID-19. In current form, the work seems just an extension of existing disease spread model.

It is suggested to add more cities and compare the spread in addition to disease dynamics in particular regions/cities.

R = 3 seems very high. I know this submission was made early bit as per latest UK data, R=0.7-1. It will be interesting to consider this point in experimentation.

There is a lot of unnecessary text in figures which make them heavy. For instance see Figure 1, mitigation intervention from 32 nd day. There is also another text filed “the 32nd day” with arrow pointing to 32nd day. Again, there is a dashed line for the same thing. These things should be avoided. It is understood and not need to add such text/arrows/lines for the same thing.

More details of statistical or mathematical model are needed. Authors may

add more parameters. It will be interesting to run experiment for major cities. The work is interesting but the scope is very limited.

6. PLOS authors have the option to publish the peer review history of their article (what does this mean?). If published, this will include your full peer review and any attached files.

Reviewer #1: No

Reviewer #2: No

Reviewer #3: No

---

## [Author Response · Author response to Decision Letter 0]

8 Jul 2020

Dear Editor and Reviewers:

First of all, we want to extend our appreciation for taking the time and effort necessary to provide such helpful and highly detailed guidance. In revising the paper, we have carefully considered your comments and endeavoured to address all of them. 

The revision includes two parts: first we summarise the major changes, then we reply to each comment in point-by-point fashion.

I Major Changes

The following changes are the most important ones:

 In the first section, we have outlined the innovative and rational explanations.

 Corrected mistakes and ambiguities in our mathematical model.

 Use our model to make predictions in more regions.

 We added simulation and description of the basic regeneration number of each intervention

 Other revisions 

 The spelling mistakes are corrected.

 The extra text and arrows of the picture have been modified. 

 Corrected where the mathematical model interpretation is ambiguous or wrong.

II Feedbacks in detail

-Reviewer 1

· the paper is NOT novel in this, either in the specific ideas proposed for covid19 or the simulations. Multiple papers have been published prior to this paper (also prior to the pre-print released mid April), and others on the broader ideas of switching quarantines in disease outbreaks.

Reply: thanks for the comment. Our paper is a time-response work and was online on 4th April 2020 which was prior to the listed papers. (please see the link: https://www.medrxiv.org/content/10.1101/2020.04.01.20043794v2 ) . We are the pioneer work to find that the number of confirmed cases in Wuhan was greatly underestimated. This is precisely because some of the infected patients are asymptomatic or mild, and they are cured by failing to go to the hospital for treatment and testing. As a result, the number of confirmed cases reported is far lower than the actual number of infected people, and the mortality rate is also high. Prior to this, there was no articles presented our findings. Also, we divided the infected population into mild and severe groups, so as to make a comparative analysis of the number of mortalities, timing and intensity of the intervention measures.

· The authors propose a 'novel' addition in Section 2 where people can move from Exposed directly to Recovered if they have no symptoms, but there is no evidence provided of examples in the published literature that these people do not move through an infectious stage before being Recovered, which would invalidate some of the model findings. The idea of asymptomatic infectious is better modelled in

Giordano, G., Blanchini, F., Bruno, R., Colaneri, P., Di Filippo, A., Di Matteo, A., Colaneri, M., and the COVID19 IRCCS San Matteo Pavia Task Force, A SIDARTHE Model of COVID-19 Epidemic in Italy, 2020.

Reply: thanks for the comment. This is a very valuable suggestion. We found in the paper [1] below, the model is defined as SIDARTHE. The E (infected but asymptomatic people) proposed in our paper actually includes the I(the virus that has been infected but has no symptoms and has not been detected) and D(asymptomatic infected, detected) proposed in the paper [1].In paper [1] has confirmed cases I (the virus that has been infected but has no symptoms and has not been detected) and D(asymptomatic infected, detected) can directly recover go to H(recovered), corresponding to the conversion of E (infected but asymptomatic people) in our paper into R(recovered).

[1] G. Giordano et al., "Modelling the COVID-19 epidemic and implementation of population-wide interventions in Italy," (in English), Nat Med, Apr 22 2020.

· The paper has very frequent errors in the English, in the text and figures. Too many examples to enumerate in the review.

Reply: thanks for the comment. We have corrected all grammatical and spelling errors by a native English speaker. 

· There is an apparent error in equation (3), should it not be I(t) = M(t)+C(t) rather than dI(t)/dt as I is just a container for M and C? or dI(t)/dt = dM(t)/dt + dC(t)/dt? There is an apparent error in equation (12) should it not be I_t = M_t+C_t rather than I_t+1 = M_t+C_t.

Reply: thanks for the comment. We have modified our formula.

(dI(t))/dt=(dM(t))/dt+(dC(t))/dt (3)

 I_(t+1)=M_(t+1)+C_(t+1) (12)

· So I think before being publishable the authors need to fix some of the model description errors, better justify the assumptions around asymptomatics and relate their approach to prior publications (long term and recent) in the field. Also they really need to clarify what extra benefit we get from their simulations compared to the previously published ones which often used more advanced models to test their simulations, and had more rigorous sensitivity analysis.

Reply: thanks for the comment. We have revised it accordingly as blow.

We estimated changes in COVID-19 transmissibility over time via the effective reproduction number(Rt) , which represents the mean number of secondary infections that result from a primary case of infection at time t. Values of Rt exceeding 1 indicate that the epidemic will tend to grow, whereas values below 1 indicate that the epidemic will tend to decline. We estimated the time-varying reproduction numbers from serial intervals and incidence of COVID-19 cases over time. 

In practical cases, it also needs to estimate the defined parameters including α_1, α_2,β, and γ_1, γ_2, γ_3, b, where β is the product of the people exposed to each day by confirmed infected people (k) and the probability of transmission (b) when exposed (i.e., β=Rt/γ= kb ) and σ is the incubation rate which is the rate of latent individuals becoming symptomatic (average duration of incubation is 1/α_1). According to recent report [20], the incubation period of COVID-19 was reported to be between 1 to 14 days, we chose the midpoint of 6 days. Preliminary data suggests that the time period α_2 from onset to the development of severe disease, including hypoxia, is 1 week [20]. γ is the average rate of recovery or death in infected populations. Using epidemic data from [6], we used SEMCR model to determine the probability of transmission (b) which was used to derive β and the probability of recovery or death (γ). The number of people who stay susceptible in each region was similar to that of its total resident population. Other transmission parameters were estimated with early prediction of Hubei cases in [6] on January 23 2020 using Monte Carlo simulation, as shown in the Table.1

Table 1: Parameters estimation in SEMCR model

Name Representation Value

N Total number of population in a region N/A

β1 Transmission rate for the I to S 0.157

β2 Transmission rate for the E to S 0.787

α_1 Incubation period 6

α_2 Incubation period from M to C 7

 γ_1 Average period from E to R 0.283

γ_2 Average period from M to R 7

γ_3 Average period from C to R 14

d Average period from C to D 28

m Mild proportion 0.8

s Severe proportion 0.138

c Critical proportion 0.061

Mt Intervention intensity 3-15

Above analysis demonstrates the complexity of controlling COVID-19 outbreaks that how and when to take what level of interventions. In this paper, we proposed a mathematical model: SEMCR to study this problem. The model extended traditional SEIR (Susceptible-Exposed-Infectious-Recovered) model [3] [4] by adding one important fact: there has been a direct link between Exposed and Recovered population. We found that the number of confirmed diagnoses reported by governments of various countries is actual much smaller than the actual number of infected persons. Although many exposed people have been infected, they will not show symptoms and transformed into infected people after the incubation period of the virus, and directly become recovered people after a certain self-healing cycle [25]. Because they are asymptomatic, they will not take the initiative to go to the hospital for treatment testing, so they will not be counted as the actual number of infected people. If we make a statistic on the cumulative number of cured people and the cumulative number of infected people, the difference between them is the number of asymptomatic patients who are directly transformed into cured people without going through the infection stage

There are two enhanced features in our model in comparison to popular SEIR models [6] [8] [12]. The first one is a straightforward relationship between Exposed and Recovered population. We find that in the early outbreaks of COVID-19, some portion of exposed people may have no obvious symptoms or only develop as mild cases, but they cannot get a test due to lack of testing kits. This group of populations might be self-recovered in some days, but will not realize they were infected. And we can calculate the approximate proportion of the number of asymptomatic or mild symptoms in the total number of infected people by counting the cumulative number of each population. The second feature in our model is that we separate infectious population into mild and critical cases in light of their symptoms. According to the curve of the number of critical cases and the number of deaths, we can make a certain explanation for the relatively high mortality rate of early outbreaks of COVID-19 after Wuhan City took immediate inhibitory interventions on January 23, 2020.Most likely, suppression intervention has led to fear in Wuhan citizens and chaos in hospital systems, as a result of delaying rescue of many critical cases. Therefore, in the next step, we can conduct a comparative analysis of the number of available medical resources and the number of critical cases in need of treatment.

 While above methods demonstrate good performance in prediction of COVID-19 outbreak by taking strong public intervention, also named as suppression strategy [13] that aims to reverse epidemic growth, one important challenge is that taking suppression strategy only is to treat disease controls as single-objective optimization of reducing the overall infectious populations as soon as possible, and require strategic consistency in a long term. In real-world, taking public health intervention strategies is actually a multiple-objective optimization problem including economic loss and society impacts. Thus, most countries have taken different intervention strategies, like enhanced surveillance and isolation to affected individuals in Singapore [14], four-stage response plan of the UK [15], mitigation approaches [13] and even multiple interventions taken in many EU countries [21-24].Due to the fact that standalone intervention strategy has apparent merits and limitations, it becomes highly necessary to study the feasibility of intervention strategies to certain country in light of its multiple natures and capabilities. 

[21] Liu, X., and Stechlinkski, P., Infectious Disease Modeling - A Hybrid System Approach, Nonlinear Systems and Complexity, Series Editor: Albert C.J. Luo, Springer Nature, 2017.

[22] X., Liu and P., Stechlinski, The Switched SIR Model. Infectious Disease Modeling, Nonlinear Systems and Complexity, vol 19. Springer, Cham, 2017. 

[23] Bin, M., Cheung, P., Crisostomi, E., Ferraro, P., Lhachemi, H., Murray-Smith, R., Myant, C., Parisini, T., Shorten, R., Stein, S. and Stone, L. 2020. On Fast Multi-Shot COVID-19 Interventions for Post Lock-Down Mitigation. (2020), 1–18.

[24] Meidan, D., Schulamann, N., Cohen, R., Haber, S., Yaniv, E., Sarid, R. and Barzel, B. 2020. Alternating quarantine for sustainable mitigation of COVID-19. i (2020), 1–29.

And we newly added the simulation and change trend description of the basic regeneration number R under various intervention measures: 

Effectiveness of Suppression Intervention: After taking intensive suppression on 23rd March in the UK, the change trend of the basic regeneration number in the UK and Wuhan is consistent. A rapid decline in R has occurred in later March, from 2.81[1.16-5.19] at the 24th day (1st March 2020) to 0.68[0.58-0.79] at the 51st day (28th March 2020). It implied implementing suppression in the UK performed significantly impact on reduction of infections. 

Figure 6: (a, c) Wuhan and (b, d, e, f) London by taking suppression intervention. Note that a, b ¬; c, d, e and f have different scales. And the orange dotted line on the X axis represents the date of implementation of the intervention, a, c is the 32nd day, b, d, e, f is the 35th day.

Effectiveness of Mitigation Intervention :The simulation of basic regeneration numbers(R) compared to suppression strategy, mitigation strategy taken in the UK gave a slower decline in R in March, from 2.73[0.97-5.40] on the 24th day (1st March 2020) to 0.98[95% CI 0.88-1.09] on the 110th day (27th May 2020). It implied that before R drops below 1, there were still much growth of infections in the UK. And it can be seen from the figure.7 that the more relaxed the intervention, the longer it takes R to fall below. 

Effectiveness of Hybrid intervention :The simulation of basic regeneration numbers(R) under the hybrid intervention in the UK shows that a similar early trend of R as suppression, where there was a fast decline in R in March, from 1.98[1.57-2.43] on the 24th day (1st March 2020) to 0.86[0.73-0.99] on the 51th day (28th March 2020). It implied that 3 weeks rolling intervention (M = 3 or 6) had equivalent effects on controlling transmissions as suppression, but need to be maintained in a longer period of 300 days. From then, R value was oscillated between 1.12 [0.95-1.29] and 0.86[0.71-1.02] with the shrinkage of intervention intensity.

Figure 7: (a, c) UK by taking Mitigation or (b, d) Hybrid intervention. Note that a, b ¬; c and d have different scales. And the orange dotted line on the X axis represents the date of implementation of the intervention, a, b, c, d is the 35th day.

-Reviewer 2

· The work is very straight forward and its hard to see any significance. Comparing London with Wuhan seems not a good fit. Both cities have different culture and life style. Similarly, Wuhan was the first city hit by corona and many things were not known by then. London had ample time with more information (spread rate, incubation period) to deal with COVID-19 disease.

Reply: thanks for the comment. different culture and lifestyles are the good points. However, regarding the epidemic per ce, the infectiousness and cure rate in all humans are similar. Also, both Wuhan and London have adopted strong isolation measures to alleviate the outbreak. Under the intervention measures of similar intensity, different cultures or lifestyles in different regions may not affect the infection rate, so we conducted a qualitative comparative analysis of these two cities.

Lastly, one limitation of our model is that its prediction of infections and deaths depends on a parameter estimation of intervention intensity that presented by average-number contacts with susceptible individuals as infectious individuals in a certain region. We assume that each intervention has the same effect on the reproduction number in different regions over time. The measures of suppression intervention in different countries and regions are similar, so the culture or other issues of different countries will not change the impact of suppression intervention on the basic regeneration number.As implementing hybrid intervention, the policy needs to be specific and well-estimated at each day according to the number of confirmed cases, deaths, mortality ratio, health resources, etc.

· In the model, it is assumed that go R directly. It seems in covid-19 this assumption is not invalid. Even if someone gets recovered, he/she gets infected. May be the justification could be that no symptoms are shown and one recovered. However, still the person can infect others without showing symptoms. This point need discussion or any references from literature.

Reply: thanks for the comment. Our model does not directly delete the R population. Our assumption for the R population is that he/she has been cured and no longer infectious. Because the reinfection rate value appears negligible. The literature(G. Giordano et al., "Modelling the COVID-19 epidemic and implementation of population-wide interventions in Italy," (in English), Nat Med, Apr 22 2020) proves our hypothesis.

Figure 2: Extended SEMCR model structure: The population is divided into the following six classes: susceptible, exposed (and not yet symptomatic), infectious (symptomatic), mild (mild or moderate symptom), critical (sever symptom), death and recovered (ie, isolated, recovered, or otherwise non-infectious).

· We find that in the early outbreaks of COVID-19, some portion of exposed people may have no obvious symptoms or only develop as mild cases, but they cannot get a test due to lack of testing kits. This group of populations might be self-recovered in some days, but will not realize they were infected". The same is true even today, You can consider undiagnosed cases too.

Reply: thanks for the comment. We’ve considered the untested cases, however, due to the insufficient data, we are not able to implement in our model, but will consider it in the further work when the data are available.

· In many figures, 10 ^ is used. Proper units such as million for x-axis should be used in figures i.e, Thousand, million etc. 10 powers are hard to understand.

Reply: thanks for the comment. We have corrected this problem in all figures, using 10^3 and 10^6 as orders of magnitude

Figure 1: Illustration of controlling Wuhan COVID-19 outbreaks by taking different intervention strategies with parameters (City populations: 1.4 million; daily contacts per person: (a)No Interventions (15), (b)Mitigation Intervention(8), (c)Suppression Intervention (3), and (d)Hybrid intervention (8-3-8-3) every two weeks; (e)Contain phase (98% effectiveness of surveillance and isolation )). Note that a, b ¬; c, d and e have different scales. The orange dotted line on the X axis represents the date of implementation of the intervention, b, c, d is the 32nd day. e is the 2nd day.

Figure 6: (a, c) Wuhan and (b, d, e, f) London by taking suppression intervention. Note that a, b ¬; c, d, e and f have different scales. And the orange dotted line on the X axis represents the date of implementation of the intervention, a, c is the 32nd day, b, d, e, f is the 35th day.

Figure 7: (a, c) UK by taking Mitigation or (b, d) Hybrid intervention. Note that a, b ¬; c and d have different scales. And the orange dotted line on the X axis represents the date of implementation of the intervention. a, b, c, d is the 35th day.

· Abstract is too long, around 450 words.

Reply: thanks for the comment. Abstract we have made reductions. The following is our revised abstract:

Recent outbreaks of coronavirus disease 2019 (COVID-19) has led a global pandemic cross the world. Most countries took two main interventions: suppression like immediate lockdown cities at epicenter or mitigation that slows down but not stopping epidemic for reducing peak healthcare demand. Both strategies have their apparent merits and limitations; it becomes extremely hard to conduct one intervention as the most feasible way to all countries. Targeting at this problem, this paper conducted a feasibility study by defining a mathematical model named SEMCR, it extended traditional SEIR (Susceptible-Exposed-Infectious-Recovered) model by adding two key features: a direct connection between Exposed and Recovered populations and separating infections into mild and critical cases. It defined parameters to classify two stages of COVID-19 control: active contain by isolation of cases and contacts, passive contain by suppression or mitigation. The model was fitted and evaluated with public dataset containing daily number of confirmed active cases including Wuhan and London during January 2020 and March 2020. The simulated results showed that 1) Immediate suppression taken in Wuhan significantly reduced the total exposed and infectious populations, but it has to be consistently maintained at least 90 days (by the middle of April 2020). Its success heavily relied on sufficiently external support from other places of China. This mode was not suitable to other countries that have no sufficient health resources. 2) In London, it is possible to take a hybrid intervention of suppression and mitigation for every 2 or 3 weeks over a longer period to balance the total infections and economic loss. While the total infectious populations in this scenario would be possibly 2 times than the one taking suppression, economic loss and recovery of London would be less affected. 3) Both in Wuhan and London cases, one important issue of fitting practical data was that there were a large portion (probably 62.9% in Wuhan) of self-recovered populations that were asymptomatic or mild symptomatic. These people might think they have been healthy at home and did not go to hospital for COVID-19 tests. Early release of intervention intensity potentially increased a risk of the second outbreak. 

· Take R0 of U, and then London.For clear understanding, you can compare figures for London “ with intervention” and "without intervention “ side by side.

Reply: thanks for the comment. We added simulation and description of the basic regeneration number of each intervention. Because our articles are written in accordance with each intervention, it may not be convenient to compare figures for London “with intervention” and "without intervention “side by side. The following is our simulation and change description of R:

Effectiveness of Suppression Intervention: After taking intensive suppression on 23rd March in the UK, the change trend of the basic regeneration number in the UK and Wuhan is consistent. A rapid decline in R has occurred in later March, from 2.81[1.16-5.19] at the 24th day (1st March 2020) to 0.68[0.58-0.79] at the 51st day (28th March 2020). It implied implementing suppression in the UK performed significantly impact on reduction of infections. 

Figure 6: (a, c) Wuhan and (b, d, e, f) London by taking suppression intervention. Note that a, b ¬; c, d, e and f have different scales. And the orange dotted line on the X axis represents the date of implementation of the intervention, a, c is the 32nd day, b, d, e, f is the 35th day.

Effectiveness of Mitigation Intervention :The simulation of basic regeneration numbers(R) compared to suppression strategy, mitigation strategy taken in the UK gave a slower decline in R in March, from 2.73[0.97-5.40] on the 24th day (1st March 2020) to 0.98[95% CI 0.88-1.09] on the 110th day (27th May 2020). It implied that before R drops below 1, there were still much growth of infections in the UK. And it can be seen from the figure.7 that the more relaxed the intervention, the longer it takes R to fall below. 

Effectiveness of Hybrid intervention :The simulation of basic regeneration numbers(R) under the hybrid intervention in the UK shows that a similar early trend of R as suppression, where there was a fast decline in R in March, from 1.98[1.57-2.43] on the 24th day (1st March 2020) to 0.86[0.73-0.99] on the 51th day (28th March 2020). It implied that 3 weeks rolling intervention (M = 3 or 6) had equivalent effects on controlling transmissions as suppression, but need to be maintained in a longer period of 300 days. From then, R value was oscillated between 1.12 [0.95-1.29] and 0.86[0.71-1.02] with the shrinkage of intervention intensity.

Figure 7: (a, c) UK by taking Mitigation or (b, d) Hybrid intervention. Note that a, b ¬; c and d have different scales. And the orange dotted line on the X axis represents the date of implementation of the intervention. a, b, c, d is the 35th day.

-Reviewer 3

· It is suggested to add more cities and compare the spread in addition to disease dynamics in particular regions/cities.

Reply: thanks for the comment. We have added Table 3 to predict detailed forecast data (total infected population, total dead population and mortality rate) of Italy, Spain, France, Belgium, Germany and Switzerland.

We used our model to estimate the impacts of suppression on controlling infections of other 6 EU countries (Italy,Spain, Germany, France, Belgium and Switzerland), as shown in Taable.3. Most suppression in other countries began around 10th-17th March (the 28th– 40thday from first confirmed case). For each country, we model the number of infections, the number of deaths, and R, the effective reproduction number over time. Specific interventions are assumed to have the same relative impact on R in each country when they were introduced there and are informed by mortality data across all countries.

As shown in Table.3, in Italy from 8th February 2020, the total number of infections is about 1.9 million, the total number of deaths is about 41 thousand, the true mortality rate is 2.17%. In Spain from 1st February 2020, the total number of infections is 2 million, the number of deaths is 41 thousand, the true mortality rate is 2.07%. In Germany from 11th February, the total number of infections is about 0.77 million, and the number of deaths is about 11 thousand. The true mortality rate is 1.5%. In France from 15th February 2020, the total number of infections is about 1.6 million, and the number of deaths is about 33 thousand. The true mortality rate is 2.09%. In Belgium from 15th February 2020, the total number of infections is about 0.62 million and the number of deaths is about 12 thousand. The true mortality rate is 1.93%, and. lastly, in Switzerland 19th from February 2020, the total number of infections is 0.23 million people, the number of deaths is about 3.4 thousand people, the real mortality rate is 1.47%. 

The time of intervention in these six European countries is similar, but from the analysis of the infections in these six European countries can be seen that the intervention effects of Germany, Belgium and Switzerland are better, and the mortality rate of Germany and Switzerland is less than Four other countries. From the data we collected can be seen that Germany and Switzerland have adequate medical resources, while Italy, Spain, France and Belgium have insufficient medical resources. This corresponds to the difference between the number of deaths and mortality we predicted. The above also confirms the previously mentioned point of view. When medical resources are insufficient, premature intervention may cause strain on medical resources, leading to more deaths and higher mortality.

Table 3: Forecasts for six other European countries

Countries Total infectious population Total dead population Mortality rate

Italy 1.9 million 41 thousand 2.17%

Spain 2 million 41 thousand 2.07%

Germany 0.77 million 11 thousand 1.5%

France 1.6 million 33 thousand 2.09%

Belgium 0.62 million 12 thousand 1.93%

Switzerland 0.23 million 3.4 thousand 1.47%

· R = 3 seems very high. I know this submission was made early bit as per latest UK data, R=0.7-1. It will be interesting to consider this point in experimentation.

Reply: thanks for the comment. We have made corrections to our R value.

Thirdly, we also find out that while COVID-19 is estimated as a high production rate (R = 2-2.5) [20], experimental evaluation results show that in either Wuhan or London cases fitted with real data in the last 6 weeks, high percentage of exposed or infectious population (at least 62.9% of infectious population in Wuhan) are actually self-recovered. These people may have no or mild symptoms but been not checked as confirmed cases. This is one important issue that Zhong’s SERI model [11] has ignored. It will answer the model [8] predicts practical infectious population in Wuhan that ten times over figures in [11]. Similarly, it could explain the estimation of practical mortality ratio can be varied in [11] and [8].

[20] Aylward, Bruce (WHO); Liang, W. (PRC). Report of the WHO-China Joint Mission on Coronavirus Disease 2019 (COVID-19). WHO-China Jt. Mission Coronavirus Dis. 2019 2019, 16–24 (2020).

· There is a lot of unnecessary text in figures which make them heavy. For instance see Figure 1, mitigation intervention from 32 nd day. There is also another text filed “the 32nd day” with arrow pointing to 32nd day. Again, there is a dashed line for the same thing. These things should be avoided. It is understood and not need to add such text/arrows/lines for the same thing.

Reply: thanks for the comment. We have made the illustrations correct, removed their heavy text and arrows, and marked the pictures with serial numbers (a, b, c, etc.), and also added detailed explanations in the explanation section.

Figure 1: Illustration of controlling Wuhan COVID-19 outbreaks by taking different intervention strategies with parameters (City populations: 1.4 million; daily contacts per person: (a)No Interventions (15), (b)Mitigation Intervention(8), (c)Suppression Intervention (3), and (d)Hybrid intervention (8-3-8-3) every two weeks; (e)Contain phase (98% effectiveness of surveillance and isolation )). Note that a, b ¬; c, d and e have different scales. The orange dotted line on the X axis represents the date of implementation of the intervention, b, c, d is the 32nd day. e is the 2nd day.

Figure 6: (a, c) Wuhan and (b, d, e, f) London by taking suppression intervention. Note that a, b ¬; c, d, e and f have different scales. And the orange dotted line on the X axis represents the date of implementation of the intervention, a, c is the 32nd day, b, d, e, f is the 35th day.

Figure 7: (a, c) UK by taking Mitigation or (b, d) Hybrid intervention. Note that a, b ¬; c and d have different scales. And the orange dotted line on the X axis represents the date of implementation of the intervention. a, b, c, d is the 35th day.

· More details of statistical or mathematical model are needed. Authors may

add more parameters. It will be interesting to run experiment for major cities. The work is interesting but the scope is very limited.

Reply: thanks for the comment. In the article, we made a very detailed description of the statistical or mathematical model, and listed detailed formulas, and also made a table to intuitively express the various parameters of the statistical or mathematical model. And we put these predictions in more regions.

We estimated changes in COVID-19 transmissibility over time via the effective reproduction number(Rt) , which represents the mean number of secondary infections that result from a primary case of infection at time t. Values of Rt exceeding 1 indicate that the epidemic will tend to grow, whereas values below 1 indicate that the epidemic will tend to decline. We estimated the time-varying reproduction numbers from serial intervals and incidence of COVID-19 cases over time. 

In practical cases, it also needs to estimate the defined parameters including α_1, α_2,β, and γ_1, γ_2, γ_3, b, where β is the product of the people exposed to each day by confirmed infected people (k) and the probability of transmission (b) when exposed (i.e., β=Rt/γ= kb ) and σ is the incubation rate which is the rate of latent individuals becoming symptomatic (average duration of incubation is 1/α_1). According to recent report [20], the incubation period of COVID-19 was reported to be between 1 to 14 days, we chose the midpoint of 6 days. Preliminary data suggests that the time period α_2 from onset to the development of severe disease, including hypoxia, is 1 week [20]. γ is the average rate of recovery or death in infected populations. Using epidemic data from [6], we used SEMCR model to determine the probability of transmission (b) which was used to derive β and the probability of recovery or death (γ). The number of people who stay susceptible in each region was similar to that of its total resident population. Other transmission parameters were estimated with early prediction of Hubei cases in [6] on January 23 2020 using Monte Carlo simulation, as shown in the Table.1 

Table 1: Parameters estimation in SEMCR model

Name Representation Value

N Total number of population in a region N/A

β1 Transmission rate for the I to S 0.157

β2 Transmission rate for the E to S 0.787

α_1 Incubation period 6

α_2 Incubation period from M to C 7

 γ_1 Average period from E to R 0.283

γ_2 Average period from M to R 7

γ_3 Average period from C to R 14

d Average period from C to D 28

m Mild proportion 0.8

s Severe proportion 0.138

c Critical proportion 0.061

Mt Intervention intensity 3-15

And we newly added the simulation and change trend description of the basic regeneration number R under various intervention measures: 

Effectiveness of Suppression Intervention: After taking intensive suppression on 23rd March in the UK, the change trend of the basic regeneration number in the UK and Wuhan is consistent. A rapid decline in R has occurred in later March, from 2.81[1.16-5.19] at the 24th day (1st March 2020) to 0.68[0.58-0.79] at the 51st day (28th March 2020). It implied implementing suppression in the UK performed significantly impact on reduction of infections. 

Figure 6: (a, c) Wuhan and (b, d, e, f) London by taking suppression intervention. Note that a, b ¬; c, d, e and f have different scales. And the orange dotted line on the X axis represents the date of implementation of theintervention, a, c is the 32nd day, b, d, e, f is the 35th day.

Effectiveness of Mitigation Intervention :The simulation of basic regeneration numbers(R) compared to suppression strategy, mitigation strategy taken in the UK gave a slower decline in R in March, from 2.73[0.97-5.40] on the 24th day (1st March 2020) to 0.98[95% CI 0.88-1.09] on the 110th day (27th May 2020). It implied that before R drops below 1, there were still much growth of infections in the UK. And it can be seen from the figure.7 that the more relaxed the intervention, the longer it takes R to fall below. 

Effectiveness of Hybrid intervention :The simulation of basic regeneration numbers(R) under the hybrid intervention in the UK shows that a similar early trend of R as suppression, where there was a fast decline in R in March, from 1.98[1.57-2.43] on the 24th day (1st March 2020) to 0.86[0.73-0.99] on the 51th day (28th March 2020). It implied that 3 weeks rolling intervention (M = 3 or 6) had equivalent effects on controlling transmissions as suppression, but need to be maintained in a longer period of 300 days. From then, R value was oscillated between 1.12 [0.95-1.29] and 0.86[0.71-1.02] with the shrinkage of intervention intensity.

Figure 7: (a, c) UK by taking Mitigation or (b, d) Hybrid intervention. Note that a, b ¬; c and d have different scales. And the orange dotted line on the X axis represents the date of implementation of the intervention. a, b, c, d is the 35th day.

---

## [Editor Report · Decision Letter 1]

16 Jul 2020

Feasibility Study of Mitigation and Suppression Strategies for Controlling COVID-19 Outbreaks in London and Wuhan

PONE-D-20-10226R1

Dear Dr. Yang,

We’re pleased to inform you that your manuscript has been judged scientifically suitable for publication and will be formally accepted for publication once it meets all outstanding technical requirements.

Kind regards,

Abdallah M. Samy, PhD

Academic Editor

PLOS ONE

---

## [Editor Report · Acceptance letter]

29 Jul 2020

PONE-D-20-10226R1 

Feasibility Study of Mitigation and Suppression Strategies for Controlling COVID-19 Outbreaks in London and Wuhan 

Dear Dr. Yang:

I'm pleased to inform you that your manuscript has been deemed suitable for publication in PLOS ONE. Congratulations! Your manuscript is now with our production department. 

Kind regards, 

on behalf of

Dr. Abdallah M. Samy 

Academic Editor

PLOS ONE